# Regulation of ZEB1 Function and Molecular Associations in Tumor Progression and Metastasis

**DOI:** 10.3390/cancers14081864

**Published:** 2022-04-07

**Authors:** Mabel Perez-Oquendo, Don L. Gibbons

**Affiliations:** 1Department of Thoracic/Head and Neck Medical Oncology, The University of Texas MD Anderson Cancer Center, Houston, TX 77030, USA; mgperez2@mdanderson.org; 2The University of Texas MD Anderson Cancer Center UTHealth Graduate School of Biomedical Sciences, Houston, TX 77030, USA; 3Department of Molecular and Cellular Oncology, The University of Texas MD Anderson Cancer Center, Houston, TX 77030, USA

**Keywords:** ZEB1, post-translational modifications, epithelial-to-mesenchymal transition (EMT), metastasis, epigenetic, transcription factor

## Abstract

**Simple Summary:**

Given the importance of the epithelial-to-mesenchymal transition in tumor progression and the pivotal role of ZEB1 as a transcriptional repressor during this process, the regulation of ZEB1-targeted gene expression is an active area of investigation. Diverse signaling pathways converge to induce ZEB1 activity, but few studies have focused on the enzyme-dependent modifications of ZEB1 that occur after its translation from mRNA (i.e., post-translational modifications, PTM). In addition to outlining the current knowledge in the field, we outline several questions regarding the PTM-mediated regulation of ZEB1 that remain unanswered. Thus, the areas of research covered in this review paper will provide a noteworthy conceptual advancement in our understanding of ZEB1′s biological function, as well as its post-transcriptional and post-translational regulation. Furthermore, the review will enhance the development of treatment strategies by identifying knowledge gaps in ZEB1′s regulatory mechanisms that could potentially be targeted to prevent and treat metastasis in cancer patients.

**Abstract:**

Zinc finger E-box binding homeobox 1 (ZEB1) is a pleiotropic transcription factor frequently expressed in carcinomas. ZEB1 orchestrates the transcription of genes in the control of several key developmental processes and tumor metastasis via the epithelial-to-mesenchymal transition (EMT). The biological function of ZEB1 is regulated through pathways that influence its transcription and post-transcriptional mechanisms. Diverse signaling pathways converge to induce ZEB1 activity; however, only a few studies have focused on the molecular associations or functional changes of ZEB1 by post-translational modifications (PTMs). Due to the robust effect of ZEB1 as a transcription repressor of epithelial genes during EMT, the contribution of PTMs in the regulation of ZEB1-targeted gene expression is an active area of investigation. Herein, we review the pivotal roles that phosphorylation, acetylation, ubiquitination, sumoylation, and other modifications have in regulating the molecular associations and behavior of ZEB1. We also outline several questions regarding the PTM-mediated regulation of ZEB1 that remain unanswered. The areas of research covered in this review are contributing to new treatment strategies for cancer by improving our mechanistic understanding of ZEB1-mediated EMT.

## 1. Introduction

Cancer is the second leading cause of mortality in the United States [1]. Metastasis is the most common cause of mortality among cancer patients [2]. Because hundreds of genes regulate the phenotype of tumor cells, there are significant barriers to identifying the modifications that initiate cancer (driver mutations) and those that respond to this initiation (passenger mutations). In addition to genetic alterations, epigenetic mechanisms of transcriptional regulation are critical to the changes that drive tumor cells to metastasize [3,4,5]

The transcription factor ZEB1 is a dominant regulator of EMT. ZEB1 is expressed in a subpopulation of cancer cells and affects their invasion and interactions with the tumor microenvironment, including immune cells that conduct surveillance [6]. High ZEB1 expression is correlated with poor outcomes, including chemotherapy resistance [7], immune suppression [8], and metastasis [9]. ZEB1 recognizes and binds to the E-boxes of epithelial gene promoter regions to suppress their transcription. Some key epithelial genes include the *CDH1* gene, which codes for E-cadherin and acts at cell–cell adherens junctions, and the *miR-200* family genes, which both repress and are repressed by ZEB1 to create a double-negative feedback loop [10,11]. ZEB1-mediated EMT also regulates critical tumor cell signaling pathways, such as the MAPK pathway in KRAS-mutant tumors [12]. ZEB1 acts through a variety of transcriptional cofactors to either repress [13,14,15] or activate [16] transcription.

Transcriptional and translational regulations contribute to the overall effect of ZEB1 in tumor progression and metastasis [17,18]. However, only a few studies have focused on the functional changes of the metastatic progression by post-translational modifications (PTMs) of the protein. PTMs refer to the enzyme-dependent modification of proteins after translation from mRNA, such as phosphorylation, acetylation, ubiquitination, and sumoylation. PTMs can notably influence the ZEB1 protein half-life, sub-cellular localization, and DNA/protein binding ability, but the data on PTM of ZEB1 have been controversial [19]. The molecular basis of cell invasiveness and migration needs to be comprehensively investigated due to the potential opposing effects of PTM-mediated molecular mechanisms. Advances in this area will not only shed light on basic mechanisms of phenotypic regulation, but will also contribute to the improvement of novel therapeutic strategies targeting PTMs.

In this review paper, we outline the biological roles of ZEB1 in mediating tumor progression and metastasis and describe the ways in which protein modifications can regulate its molecular associations and function. The work described herein underscores the need to further interrogate the biomolecular basis of PTM-mediated regulation of ZEB1′s activity, stability, localization, and protein–protein interactions.

## 2. Biological Function of ZEB1

### 2.1. Structure of ZEB1

ZEB1 (originally called TCF8 or δEF1) [20] is a member of the zinc finger and homeobox transcription factor family and is positioned on chromosome 10p11.2 [21]. The ZEB1 protein contains 1117 amino acids, including the C2H2-type flanking zinc finger amino-terminal cluster (NZF) and carboxy-terminal cluster (CZF) on either end and the homologous structural homeodomain (HD) in between [22] (Figure 1). The two C2H2-type zinc fingers are responsible for recognizing and binding the specific 5′-CANNTG-3′ sequence [23] that overlaps with the E-boxes of gene promoter regions [24,25]. In addition, the NZF and CZF clusters regulate cell differentiation and tissue-specific functions [22,26]. The middle HD region is flanked by the Smad interaction domain (SID) and the C-terminal binding protein (CtBP)-interaction domain (CID) [27]. These interaction domains are instrumental in recruiting additional proteins with the capability to control ZEB1′s transcriptional activity [25]. Given the numerous protein associations and gene targets, ZEB1 is considered a pleiotropic transcription factor. Therefore, the regulation of ZEB1 in tumor progression is highly dependent on the cellular and tissue context.

### 2.2. ZEB1 in Tumor Progression

By targeting epithelial gene promoters, ZEB1 is a dominant regulator of tumor cells and other cell types essential to tumor progression. High ZEB1 expression is observed in multiple cancer types, including lymphoma, bladder, brain, breast, cervical, colon, endometrial, gastric, head and neck, liver, lung, pancreatic, renal, and uterine cancer [28,29,30,31]. One study of 117 patients with pancreatic ductal adenocarcinoma revealed that patient prognosis is inversely correlated with the levels of ZEB1 expression in stromal cancer-associated fibroblasts [32]. Similar observations were noticed in patients with colorectal cancer, uterine cancer, and endometrial cancer. In addition, high levels of the transcription factor ZEB1 have been found in more than 95% of patients with cervical cancer, and its expression correlates with the stages of the International Federation of Gynecology and Obstetrics [33]. Furthermore, ZEB1 expression occurs early in gastric carcinoma development to promote tumor progression and metastasis [34].

ZEB1 expression promotes tumor progression by not only enhancing cell motility through the disassociation of polarized epithelial cells but also by reprogramming the surrounding tumor microenvironment to facilitate migration and invasion. In prostate cancer, ZEB1 promotes vasculogenic mimicry through Src signaling, which is associated with mesenchymal and cancer stem cell phenotypes [35]. In mesenchymal stem cells, ZEB1-repressed miRNA clusters upregulate secreted fibronectin 1 and serine protease inhibitor family E member 2 to stimulate autocrine signaling, which is associated with increased vasculogenic mimicry in breast cancer cells [36]. In MDA-MB-231 breast cancer cells, high ZEB1 expression increases the synthesis of vascular endothelial growth factor A, thereby inducing tumorigenesis and angiogenesis [37]. Endothelial ZEB1 deletion in tumor-bearing mice diminishes tumor angiogenesis and reduces TGF-β activity, thus reducing tumor growth and metastasis. Moreover, exposure of low-dose antiprogrammed cell death protein 1 (PD-1) antibody elicits tumor regression in and extends the survival of ZEB1-deleted mice [38].

ZEB1-mediating upregulation of various inflammatory cytokines is contributing to tumor formation and growth. In breast cancer, ZEB1 promotes inflammation in the tumor microenvironment by regulating the secretion of the proinflammatory cytokines interleukin 6 (IL-6) and IL-8 and subsequently inducing the growth of fibroblasts and myeloid-derived suppressor cells [39]. Strikingly, the ZEB1/p53 axis in stromal fibroblasts promotes the development of mammary epithelial tumors. High ZEB1 expression in the stroma correlates with increased extracellular matrix remodeling, immune cell infiltration, angiogenesis arising from the increased expression of IL-6, FGF2/7, vascular endothelial growth factor, and the secretion of these factors into the surrounding stromal tissue [40]. Thus, by reforming the tumor microenvironment, ZEB1 is vital to tumor progression.

### 2.3. ZEB1 in Metastasis

Metastasis causes epithelial cells with highly specialized cell–cell contacts and defined apical–basal polarity to detach from the primary tumor and develop mesenchymal features, such as motility and invasiveness through the process termed EMT [4]. The evolutionarily conserved developmental EMT program is mediated by several transcriptional factors, one of which is ZEB1 [41]. ZEB1 recognizes and directly binds to the promoter regions of epithelial genes to suppress their transcription [23,42] and promote EMT by activating the transcription of mesenchymal genes [43,44]. In non-small cell lung cancer (NSCLC) cells, ZEB1-mediated suppression of E-cadherin can induce migration, invasion, and activation of the epidermal growth factor receptor (EGFR) [45]. In addition to E-cadherin, ZEB1 acts on the cell polarity genes *HUGL2*, *Crumbs3*, and *PATJ* to repress their transcription. In lung, breast, and colorectal cancer, this suppresses cell–cell adhesion and epithelial differentiation, which correlates with primary tumor detachment, invasion, and metastasis [46,47]. In addition to downregulating epithelial genes, ZEB1 represses basement membrane synthesis, enhances collagenous matrix production, and activates the transcription of matrix metalloproteases (MMP) 1, 9, and 14, thus promoting extracellular matrix remodeling and tumor cell invasion [10,48,49,50]. In mantle cell lymphoma, ZEB1 activates the antiapoptotic genes *MCL1*, *BCL2*, and *BIRC5* as well as the proliferative genes *HMGB2*, *UHRF1*, *CENPF*, *MYC*, *MKI67*, and *CCND1* while inhibiting the proapoptotic genes *TP53*, *BBC3*, *PMAIP1*, and *BAX* [51]. Induced ZEB1 expression is also associated with the enriched metastatic properties of xenotransplanted epithelial cancer cells [52]. Conversely, the mesenchymal-to-epithelial transition (MET) mechanism is associated with suppressed ZEB1 expression. Metastatic breast cancer cells with lower ZEB1 expression levels develop an epithelial phenotype and lose the mesenchymal/motile phenotype. Thus, understanding the regulation of ZEB1 in driving tumor progression and invasion could provide awareness of the biomolecular mechanisms through which cells acquire metastatic properties.

### 2.4. ZEB1 in Therapy Resistance

Emerging evidence indicates that during tumor differentiation, cells acquire molecular and phenotypic changes that resemble the EMT process. The association of an epithelial or mesenchymal state with therapy resistance depends on the cancer and treatment types, as well as the inducer of EMT [53]. ZEB1, through PTMs and its interaction with miRNAs, is implicated in resistance to various anticancer therapies [54]. ZEB1 upregulation stabilizes the checkpoint kinase CHK1 by activating the deubiquitylation of USP7. This in turn upregulates homologous recombination-dependent DNA damage repair (DDR) and tumor radioresistance [55]. Zhang et al. identified a mechanism of ZEB1 involvement in radiotherapy resistance. They found that mechanistically, ZEB1 is phosphorylated and stabilized by ATM after X-ray irradiation, which in turn represses the negative regulator miR-205. This repression increases ZEB1 and ubiquitin-conjugating enzyme Ubc13 levels and improves DDR. Conversely, miR-205, by targeting ZEB1 and Ubc13, impairs DDR [56]. Therefore, the authors suggested that ZEB1-targeting agents, such as miR-205 mimics, are a promising therapy for radioresistant tumors. ZEB1 expression may also promote resistance to new therapies that target major biological pathways associated with cancer. For example, targeting and suppressing ZEB1 with the HDAC1-specific inhibitor mocetinostat or through the expression of miR-200 sensitizes resistant cancer cells to MEK inhibitors and reduces tumor growth in vivo [12]. This underlines the complexity and pleiotropic role that ZEB1-targeting agents have in regulating ZEB1 response to various anticancer treatments.

## 3. Regulation of ZEB1 Expression

### 3.1. Transcriptional Regulation of ZEB1

Transcriptional regulation is defined as the regulation of the processes that directly convert DNA into RNA, thereby modulating gene expression [57], and allow the cell to respond to changing signals. Because it is downstream of several crucial transcriptional pathways, ZEB1′s involvement in the RAS/ERK [45], TGF-β [58], PI3K/Akt [59], and NFκB [60] pathways helps drive tumorigenesis and metastasis. Whether these signaling pathways are activated or inhibited dictates the course of EMT and subsequent programs in tumor cells. The ERK/MAPK pathway activates ZEB1 transcription to induce prostate cancer cell invasion and metastasis via hepatocyte growth factor [61]. The RAF/ERK pathway and the SP1 and JUN transcription are activated by the KRAS oncogene. These events promote ZEB1 function to inhibit miR-200 activity, which leads to EMT [62]. TGF-β activates ZEB1 transcription by upregulating pSmad2 expression to provoke the mesenchymal state in multiple tumor types [63]. In EGFR-mutant NSCLC, the long noncoding RNA BC087858 induces EGFR tyrosine kinase inhibitor (TKI) resistance through ZEB1-mediated activation of the MEK/ERK and PI3K/Akt pathways [64]. Interestingly, the inhibition of ZEB1 transcription facilitates caspase 3−mediated apoptosis by downregulating the NFκB/iNOS pathway, thus inhibiting the proliferation of MG-63 osteosarcoma cells [65]. The NFκB pathway increases ZEB1 transcription and induces metastasis, lymphovascular invasion, and neural invasion to promote pancreatic cancer progression [66]. Overall, ZEB1 is regulated by diverse signaling pathways at the transcriptional level, which in many cases are context- or tumor-specific.

The chromatin arrangement of ZEB1 is defined mostly by PTMs of histones that dictate the transcriptional status. The trimethylation of lysine 4 of the histone H3 subunit (H3K4me3) and dimethylation of lysine 79 of the histone H3 subunit (H3K79me2) have been associated with transcriptional initiation [67] and elongation [68], respectively. The enrichment of H3K4me3 and H3K79me2 is indicative of a gene that is actively transcribed [67]. Conversely, trimethylation of lysine 27 of the histone H3 subunit (H3K27me3) has been correlated with transcriptional gene repression [69]. The promoter of ZEB1 is associated with a chromatin region occupied by both permissive H3K4me3 and restrictive H3K27me3, better known as a bivalent chromatin domain [70]. This bivalent domain represses ZEB1 transcription, but in response to signals favoring differentiation, the transcriptional activity can switch to produce high ZEB1 levels. One investigation revealed a connection between ZEB1 as a chromatin-modifying protein and CD44 expression. Embryonic stem cells with high CD44 expression displayed chromatin methylation patterns indicative of high ZEB1 transcription levels. These findings were supported by the enrichment of both H3K4me3 and H3K79me2 and the absence of H3K27me3 at the ZEB1 promoter. However, low CD44 expression in these cells led to bivalent chromatin methylation at the ZEB1 promoter owing to the presence of both H3K4me3 and H3K27me3, but not H3K79me2 [71]. Overall, these results indicate that untransformed basal mammary epithelial cells that can shift to a stem-like form with high CD44 expression maintain a bivalent configuration at the ZEB1 promoter. This configuration enables the cells to promptly respond to changes in their microenvironment [71]. These dynamic chromatin modifications may be responsible for variances in ZEB1 expression in many cases.

### 3.2. Feedback Loop-Mediated Regulation of ZEB1 Expression

The regulation of ZEB1 through a feedback loop can mediate tumor progression and metastasis by communication among key signaling pathways. Negative feedback serves to decrease the output: a stable state is achieved by reducing the result of a reaction. However, positive feedforward pathways serve to increase the output: the reaction is achieved faster by amplifying its result [72]. Disruption in the function of key feedback loops can result in various undesirable consequences, such as ZEB1-mediated tumor progression.

ZEB1 both represses and is repressed by the miR-200 family members (miR-200a-c, miR-141, and miR-429), thereby creating a double-negative feedback loop known as the ZEB1/miR200 feedback loop [10,11]. ZEB1 directly represses the miR-200 family through its binding to miR-200 promoter regions, which in turn induces a mesenchymal, spindle cell morphology [73,74]. Conversely, the miR-200 family binds to the *ZEB1* mRNA at the 3′-untranslated region (3′-UTR), which comprises eight miR-200 family binding sites that serve to repress EMT. ZEB1/MYB forms a separate reciprocal negative feedback loop to regulate proliferation, apoptosis, and differentiation. MYB inhibits ZEB1 expression and consequently alleviates ZEB1′s transcriptional repression of *CDH1*, which promotes the expression of E-cadherin [75]. Similarly, GRHL2 and ZEB1 create a double-negative feedback loop to regulate the epithelial phenotype. The activation of GRHL2 inhibits ZEB1 activity, alleviates *CDH1* transcription, and induces the epithelial state by the miR-200 family [76]. In contrast, the mesenchymal state is achieved when ZEB1 levels are upregulated through GRHL2 knockdown. A recent study in lung adenocarcinoma revealed an inverse correlation of the anterior gradient protein 2 (AGR2) and ZEB1. High expression of ZEB1 disrupts the transcription of the *AGR2* gene, while AGR2 activity facilitates *ZEB1* mRNA degradation. The ZEB1/AGR2 double-negative feedback loop is associated with lung cancer invasion and metastasis [77].

In addition to having a role in negative feedback loops, ZEB1 is also a key player in the regulation of several positive feedforward loops. ZEB1 forms a positive feedforward loop with CD44s to maintain stem cell features, adhesion, migration, and signaling. ZEB1 inhibits the epithelial splicing regulatory protein 1 (ESRP1), triggering a change from the epithelial-specific CD44v isoform to the standard CD44s isoform, which maintains the stemness and mesenchymal characteristics of cancer cells [78]. Hyaluronic acid (HA), a major extracellular matrix proteoglycan that is synthesized mainly by hyaluronic acid synthase 2 (HAS2) [79], combines with CD44s to activate the expression of ZEB1 in tumor cells, indicating that HAS2 and ZEB1 form a positive feedforward loop. ZEB1 recognizes and binds to the E-boxes of the HAS2 promoter to foster additional HA synthesis, thereby generating a self-enforcing feedforward loop. Thus, ZEB1 can utilize HAS2/HA to orchestrate a supportive microenvironment and enforce its own expression [80]. Interestingly, the ZEB1/ESRP1/CD44 loop controls the ZEB1/HAS2/HA loop, indicating a complex multifactorial positive feedforward system [81]. The control of feedback loops in ZEB1-mediated EMT is consistent with the mechanistic linkage of ZEB1′s interaction partners and stability in a manner that invokes epigenetic regulation.

### 3.3. Epigenetic Regulation of ZEB1

Epigenetic regulation represents an important mechanism by which EMT transcription factors recognize and bind to gene promoter regions [82]. Several investigations in recent years agree that the memory of lineage-specific transcription factors with regulatory elements of gene expression is an epigenetic mechanism that maintains cellular identity [83]. RAB25 and ESRP1 are key regulators of phenylbutyrate sensitivity, a histone deacetylase antagonist that represses tumor activity in breast cancer cell lines. RAB25 and ESRP1 levels were downregulated by the administration of DNA demethylation treatments or ZEB1-regulated through epigenetic regulation [17]. Mechanistically, these genes synergistically repressed proliferation in an siRNA-induced knockdown of ZEB1, suggesting the use of ZEB1’s epigenetic regulation as a biomarker for overcoming drug resistance. The protein arginine methyltransferase 1 (PRMT1) is another key regulator of EMT epigenetic control through its transcription factor ZEB1. Silencing of PRMT1 drives cell cycle arrest and cellular senescence in vitro and suppresses metastasis in vivo. PRMT1 mediates the asymmetric dimethylation of arginine 3 of histone H4 (H4R3me2as) in ZEB1 to trigger its transcriptional activity [18]. Therefore, epigenetic targets are valuable in preventing cancer metastasis through regulation of ZEB1.

### 3.4. Translational Regulation of ZEB1

In most cases, mRNA translation plays a dominant role in treatment resistance and tumor progression [84]. The ability of the cap-binding protein eukaryotic translation initiation factor 4E (eIF4E) to direct ribosomal translation of the mRNAs is key in the process where a normal cell becomes a cancer cell. The phosphorylation of eIF4E at the Ser209 residue is mediated by MAPK-interacting protein kinases 1 (MNK1) and 2 (MNK2) and contributes to mRNA translation [85]. The heterogeneous nuclear ribonucleoprotein A1 (hnRNPA1), an MNK effector, oversees the mRNA splicing and translation. The hnRNPA1 protein has an affinity for the AUUUA sequence; notably, ZEB1 has 9 AUUUA sequences in its 3′-UTR. Experimentally, hnRNPA1 knockdown increases the production of ZEB1 protein without disturbing the levels of *ZEB1* mRNA, suggesting that hnRNPA1 can repress the translation of *ZEB1* mRNA [85]. Beclin 1 also contributes to the stability of *ZEB1* mRNA. In thyroid cancer, haploinsufficiency of Beclin 1 triggers EMT through the mRNA stabilization of ZEB1. AU-binding factor 1 (AUF1) is upregulated by the knockdown of Beclin 1, associates with the *ZEB1* mRNA at the 3′-UTR to decrease its degradation and increase its half-life [86]. AUF1 can also be upregulated by ZEB1-antisense 1 (ZEB1-AS1). Mechanistically, ZEB1-AS1 binds to AUF1, thereby activating the *ZEB1* translation without disturbing its mRNA level, causing a rise in ZEB1 protein levels [87]. Thus, the combination of various quantity control pathways (e.g., transcription, miRNA expression, and 3ʹ-UTR) and quality control pathways (e.g., mRNA decay) significantly contribute to the regulation of ZEB1 protein levels.

## 4. Post-Translational Modifications of ZEB1 

ZEB1 cooperates with other cellular regulators to suppress epithelial characteristics and induce metastasis. Diverse signaling pathways converge to activate ZEB1 and induce the mesenchymal phenotype. However, the ways in which PTMs change ZEB1′s function in tumorigenesis are unclear [19]. Studying the pivotal roles that phosphorylation, acetylation, ubiquitination, and sumoylation have in regulating ZEB1 is important because it could provide a significant conceptual understanding of an integrated network to explain its molecular associations and functional lifecycle.

### 4.1. ZEB1 Protein Stability

PTMs are important in defining protein–protein interactions and protein stability. Therefore, ZEB1 turnover has been assessed in several cancers through cycloheximide chase assays. Cycloheximide (CHX) inhibits the translocation step during ribosomal translation, resulting in impaired de novo protein synthesis [88]. ZEB1, which has a half-life of ~3 h, is ubiquitinated and targeted to be degraded by the proteasome system [89]. SIAH proteins, ubiquitin ligases (UBLs) reported as drivers of ZEB1 proteasomal degradation, regulate substrates by the transfer of ubiquitin. High expression of either SIAH 1 or 2 is correlated with a shortened ZEB1 half-life in human osteosarcoma. However, cotransfection with the UBL-deficient RING form of SIAH1 or 2 returns the ZEB1 half-life to 3 h and to more than 4 h, respectively. These UBL-deficient RING forms recognize and bind to substrates but are unable to transfer ubiquitin to substrates owing to alterations in their respective RING domains. Furthermore, SIAHs can increase ZEB1 ubiquitination rates, as they have been shown to be binding partners [89]. Caspase-8-associated protein 2 (FLASH) also protects ZEB1 from the ubiquitin–proteasome system via the E3-UBLs SIAH1 and the F-box protein FBXO45. Specifically, in the absence of FLASH, the ZEB1 half-life decreases to 1 hour in HeLa 229 cells. Interestingly, the proteasome inhibitor MG132 restores ZEB1 levels in FLASH-depleted cells, suggesting that loss of FLASH function reduces the ZEB1 protein stability by increasing ZEB1 proteasomal degradation [90]. Intriguingly, the loss of USP51 deubiquitinase decreases the half-life of endogenous ZEB1 protein, whereas MG132 induces the polyubiquitination of ZEB1 in LM2 cells (a lung metastatic subline of MDA-MB-231 cells) [91]. Thus, different PTMs make pivotal contributions to the protein stability and cellular levels of ZEB1.

### 4.2. Phosphorylation of ZEB1

Phosphorylation is a common, well-characterized modification involved in a wide-ranging of biomolecular mechanisms that drive proliferation, DNA damage, and tumor progression. Phosphorylation occurs at tyrosine, threonine, and serine residues by kinases, while phosphatases catalyze their dephosphorylation [41]. IGF-1 phosphorylates ZEB1 at Threonine-867 through the MEK/ERK pathway and leads to transcriptional repression by ZEB1. In addition, PKC represses ZEB1 through the phosphorylation of Threonine-851, Serine-852, and Serine-853 (Figure 2). Intriguingly, mutations of the Threonine-851, Serine-852, and Serine-853 triplet subsequently block IGF1/MEK/ERK signaling, which suggests that the PKC and MEK/ERK pathways occur successively to inhibit ZEB1 activity. IGF1/MEK/ERK signaling is also hindered by the Threonine-867 phosphorylation, which seems to tolerate the phosphorylation of Threonine-851, Serine-852, and Serine-853 through the PKC signaling pathway, thereby inhibiting ZEB1 activity [92]. As a whole, repression of ZEB1 activity by the MEK/ERK, PKC, or PI3K pathways balances ZEB1′s activation by TGF-β signaling and allows ZEB1 to integrate these pathways in response to the cellular microenvironment.

Whereas MEK/ERK, PKC, and PI3K downregulate ZEB1, the DNA damage-sensing kinase ataxia–telangiectasia mutated (ATM) upregulates ZEB1. Mechanistically, ATM is activated upon exposure to γ-ionizing radiation, resulting in the phosphorylation of several key regulators in the cell cycle arrest, DNA damage, and apoptosis pathways. ZEB1 physically interacts with ATM in SUM159-P2 human breast cancer cells, such that ATM depletion significantly downregulates ZEB1 and checkpoint kinase 1 (CHK1) protein levels. The ZEB1 protein has a conserved S/T-Q motif containing Serine-585, which is a common motif for ATM substrates. The ATM inhibitor Ku55933 significantly decreases the ZEB1 protein stability and reduces the S/T-Q phosphorylation. ATM can phosphorylate ZEB1 at the Serine-585 residue, which is critical for the radiation-induced ZEB1 stabilization (Figure 2) [55]. This study reveals that ZEB1 is upregulated by autophosphorylation as well as by ATM hyperactivation in radioresistant breast cancer cells.

### 4.3. Ubiquitination of ZEB1

Ubiquitin is a small protein that regulates the processes of other proteins. During ubiquitination, a ubiquitin molecule binds to a substrate protein to modulate its functionality. Ubiquitination is a reversible process with the subtraction of ubiquitin molecule(s) catalyzed by deubiquitinating enzymes (DUBs or deubiquitinases) [93]. The aberrant expression of certain DUBs is associated with many diseases, including cancer. DUBs are responsible for the stabilization of the ubiquitination rate and proteasomal degradation of ZEB1. The ubiquitin ligase FBXW7 mediates the ubiquitin-dependent proteolysis of oncoproteins, including cyclin E1, mTOR, and NOTCH. Of note, mTOR is considered a potential therapeutic target in cholangiocarcinoma (CCA), since elevated mTOR activity is positively correlated with activated EMT. Mechanistically, the mTOR inhibitor rapamycin inhibits the EMT, stem-like capacity, and migration of cholangiocarcinoma in vitro and inhibits the FBXW7 silencing-induced metastasis of these cells in vivo. Furthermore, rapamycin treatment markedly decreases ZEB1 expression, thereby inhibiting the mesenchymal features of both CCA cells and FBXW7-deficient CCA cells. These observations indicate that ZEB1 plays a role in the ubiquitin-targeted FBXW7/mTOR pathway, which is implicated in EMT [94]. In addition, USP51 deubiquitinates and stabilizes the ZEB1 protein. In the cisplatin-resistant lung cancer cell line A549/DDP, USP51 knockdown significantly induces apoptosis and the ubiquitin-mediated degradation of ZEB1, whereas overexpression of USP51 decreases the cleavage of PARP-1 and caspase-3. Notably, ZEB1 overexpression potently diminishes the effects of USP51 knockdown in these cells [95]. Collectively, these findings demonstrate that the inhibition of mTOR or USP51 attenuates the EMT phenotype through the ubiquitin-mediated degradation of ZEB1. A recent investigation evaluated and described the ubiquitination levels of ZEB1 mediated by the E3 ubiquitin ligase TRIM26 and deubiquitinating enzyme USP39. Mechanistically, TRIM26 ubiquitinates and targets ZEB1 for degradation, while USP39 deubiquitinates and stabilizes ZEB1 to promote hepatocellular carcinoma (HCC) [86]. Understanding the mechanisms that regulate ZEB1 ubiquitination levels may provide a promising strategy for targeting E3 ubiquitin ligases and deubiquitinating enzymes in the treatment of tumor progression and metastasis.

### 4.4. Sumoylation of ZEB1

The sumoylation pathway results in the reversible attachment of a small ubiquitin-like modifier (SUMO) to a lysine residue in the protein. Intriguingly, sumoylation significantly influences the function of transcription factors even when a minor portion of the protein is sumoylated [96]. Polycomb protein (Pc2) sumoylates ZEB1, thereby affecting its repressive activity. The Pc2-induced SUMO conjugation of ZEB1 at K347 and K774 diminishes ZEB1′s repression of E-cadherin (Figure 2) [97]. In HeLa cells transfected with the CtBP1-Ubc9 fusion protein, CtBP1 facilitates the sumoylation of ZEB1, thus exhibiting a change in the molecular weight of ZEB1 [98]. These findings propose that cofactors (i.e., CtBP) contribute to the modifications (i.e., sumoylation) of EMT-transcription factors, such as ZEB1, to post-translationally regulate their activity and function as transcriptional repressors.

### 4.5. ZEB1 Interactions with Histone Acetyltransferases

The acetylation of proteins is catalyzed by histone acetyltransferases, including P300, CBP, and PCAF, which contain acetyl-transfer capability of the substrate. Inversely, histone deacetylases (HDACs) remove acetyl groups to modulate protein function, interaction, and stability. Several studies have acknowledged that transcription, signaling pathways, and diverse intracellular events are controlled by the acetylation status mediated by histone acetyltransferases and HDACs [99]. The ability of ZEB1 to promote transcription at target gene promoters has been implicated with the recruitment of histone-modifying cofactors to the gene promoters. P300 and CBP, the transcriptional coactivators of ZEB1, are actively involved in countless biological processes, including proliferation, differentiation, DNA damage repair (DDR), and carcinogenesis [100]. In HuCCT-1 cholangiocarcinoma cells overexpressing SPRR2A, acetyl-transfer activity controls miR-200c transcription [101]. The acetyltransferases P300 and PCAF recognize and bind to the miR-200c/miR-141 promoter to induce promoter activation, whereas the disruption of P300 and PCAF contacts strongly suppresses the expression of miR-200c and miR-141. Furthermore, treatment with the HDAC1 inhibitor trichostatin A significantly upregulates miR-200c and miR-141 and enhances their promoter activity, suggesting that trichostatin A, as well as P300 and PCAF, can promote the cellular acquisition of epithelial characteristics. This also indicates that, in some cases, it is the acetylation of the protein that binds to the target gene promoters that enhances the promoter activity of miR-200c and miR-141.

ZEB1 has a prominent role in the regulation of bone metabolism, promoting both angiogenesis and osteogenesis. ZEB1 also associates with P300 and CBP in bone endothelial cells and co-occupies the promoters of Dll4 and NOTCH1 to foster their histone acetylation, which subsequently activates the transcription of Dll4 and NOTCH1. Interestingly, in bone endothelial cells, ZEB1 deletion does not affect *CDH1* transcription. Instead, ZEB1 deletion decreases histone acetylation on the promoters of Dll4 and NOTCH1, thus suppressing their transcription and downregulating NOTCH signaling. NOTCH signaling is a critical regulator of vascular development, migration, lumen formation, and cell differentiation [102]. Taken together, histone acyltransferases (i.e., P300) can bind to ZEB1 to affect epigenetic chromatin modeling and, therefore, transcription of ZEB1 target genes that regulate metastasis.

### 4.6. ZEB1 Binding to Co-Repressors

The foremost function of ZEB1 is to suppress the transcription of epithelial genes, such as *CDH1* and the miR-200 family, and increase the expression of mesenchymal proteins, such as vimentin. In breast cancer, ZEB1 represses the transcription of the *CDH1* gene, by targeting the E-box of the gene promoter [42]. ZEB1 binding to the *CDH1* promoter caused the recruitment of the CtBP to ZEB1’s CID, which ultimately suppresses *CDH1* transcription and promotes tumor progression and metastasis (Figure 3A). A ZEB1/CtBP complex also affects the levels of the growth hormone interleukin 2 (IL-2) and the transcription factor Bcl-6 [13,103], which are associated with the development of T regulatory cells and germinal center B cells, respectively. In addition, the repression of E-cadherin may be independent of CtBP, as ZEB1 can use the SWI/SNF chromatin-remodeling brahma-related gene-1 (BRG1) protein as a replacement corepressor for CtBP. BRG1 interacts with ZEB1 at the N-terminal region, in the invasive front of carcinomas (Figure 3A). BRG1/ZEB1 suppresses E-cadherin activity and upregulates vimentin activity, thereby contributing to the development of the mesenchymal phenotype [14]. Another CtBP-independent mechanism involves ZEB1’s suppression of its target genes by recruiting the nucleosome remodeling and deacetylase (NuRD) complex (Figure 3A). The ZEB1/CHD4/NuRD complex mediates the repression of miR-200c, miR-141, and TBC1D2b. The suppression of TBC1D2b contributes to the degradation of E-cadherin, thus enhancing invasiveness and EMT-mediated metastasis [15]. These studies exhibit how ZEB1, as an EMT-mediated transcription factor, regulates the suppression of epithelial gene transcription through the recruitment of and association with multiple other corepressor protein complexes.

### 4.7. ZEB1 Binding to Transcriptional Activators

ZEB1 binding to transcriptional activators requires association with the PC2-CtBP-LSD1-LCoR complex or the SWI/SNF chromatin-remodeling BRG1 to lead the ZEB1-Smad3-P300-P/CAF complex [100]. ZEB1 binds activated Smad and P300 proteins, which facilitates the assembly of a Smad-P300 complex while dissociating ZEB1 from its co-repressors. ZEB1 promotes the transcription of the TGF-β−responsive genes through its SID domain. Thus, in association with the Smad-P300 complex, ZEB1 can change from a transcriptional repressor to an activator of the TGF-β-responsive genes (Figure 3B) [25,104]. In addition, the effector of the Hippo/Yes-associated protein (YAP) pathway can directly switch the transcription ability of ZEB1. The interaction between YAP and ZEB1 converts ZEB1 from a transcriptional repressor to an activator that targets TEAD-binding sites. YAP interacts with ZEB1 in places known to bind P300 (NZF) and possess a transcriptional activation domain (CZF) (Figure 3B). ZEB1/YAP complex enhances the transcription of mutual ZEB1/YAP target genes, which is a predictor of therapy resistance and metastasis.

## 5. Conclusions

Post-translational modifications can orchestrate multiple activities of transcription factors, including their localization, protein complex interactions, DNA binding, protein stability, and functional transcriptional activity. Mounting evidence suggests that the PTM-mediated regulation of transcriptional factors is crucial to understanding various diseases, especially within the context of cancer progression and metastasis. Given the prominence of ZEB1 as a transcription repressor of epithelial genes during EMT, the role of PTMs in the epigenetic regulation of ZEB1-targeted gene expression is an active area of investigation. ZEB1’s activity and expression differ on a case-by-case basis. For instance, diverse post-transcriptional mechanisms, including signaling pathways and feedback loops, converge to regulate the mRNA and protein stability. However, few studies have focused on the enzyme-dependent modifications of ZEB1 that occur after its translation from mRNA (i.e., PTMs). On the basis of our own and others’ studies, we propose that post-translational mechanisms play a prominent role in defining ZEB1 interactions to regulate cell fate and tissue-specific outcomes. The roles that phosphorylation, acetylation, ubiquitination, sumoylation, and other modifications have in regulating the behavior of ZEB1 require additional investigation.

Given its numerous protein partners and target genes, ZEB1 is a pleiotropic transcription factor whose function largely depends on cellular, genetic, and epigenetic factors. Protein–protein interaction is one of the main classes of therapeutic targets for cancer. The chromatin configuration (i.e., euchromatin vs. heterochromatin) and the presence or absence of tissue-specific cofactors strongly influence the function of ZEB1 as a transcriptional repressor and/or activator. This may explain why targeting ZEB1 remains a challenge, as ZEB1 promotes complementary or synergistic effects in some cells, but apparently has opposite functions in others. Although CtBP, BRG1, and NuRD work independently as ZEB1 corepressors, each efficiently represses E-cadherin [13,14,15]. More research is warranted to determine whether the corepressor complexes differentially regulate E-cadherin or other target genes of ZEB1 across different tissues or at different stages of differentiation and development. Additionally, it will be imperative to understand how ZEB1 binding/interaction in distinct protein repressor complexes is determined, e.g., what determines whether the protein will interact with the NuRD vs. BRG1 complexes.

CBP and P300, through their acetyl-transfer activity, participate in the transcriptional regulation of ZEB1 by contributing to the formation of protein complexes. However, the contribution of P300 in tumor progression and metastasis is controversial. Some studies have shown that elevated P300 levels lead to EMT [105], whereas others have shown that null P300(^−/−^) cells also stimulate EMT [106]; the variances could be context-specific. As mentioned above, P300 forms a P300/PCAF/ZEB1 complex on the miR-200c promoter and acetylates ZEB1 to activate miR-200c. P300 siRNA treatment significantly decreases miR-200c expression but has no consequence on E-cadherin levels in epithelial (vector) cells. In mesenchymal (SPRR2a clone) cholangiocarcinoma cells, however, P300 siRNA significantly downregulates E-cadherin expression [101]. Thus, a reduction in P300 mediates the transcription of ZEB1 target genes. P300 can also serve as a transcriptional coactivator that recognize the E-cadherin promoter and enriches gene transcription. The exact mechanisms by which cells regulate P300/PCAF are unknown but likely involve a lack of cofactors characterization, limiting strategies to target ZEB1’s molecular mechanisms.

Finally, ZEB1 is predicted to have a molecular weight of 125 kDa; however, numerous groups report that ZEB1 instead exists in two different species—a 125 kDa and 190,250 kDa band by gel electrophoresis. This observation supports the hypothesis that ZEB1 is subject to extensive PTMs, including acetylation, phosphorylation, and ubiquitination. However, these modifications do not entirely account for the large inconsistency between the molecular weights of ZEB1. The current lack of structural data for ZEB1 hamper drug development to target the protein or its interactions. In addition to PTMs, dimerization is a common mechanism that can unlock novel functions, promote stability, or enhance DNA binding [107]. Transcription factors can function as monomers if the DNA-binding domain is capable of modifying the fate of transcription (activation or repression) [108]. However, because the DNA-binding domains of monomers bind to their target genes with low efficiency, transcription factors frequently undergo dimerization. It is worth noting the example of the transcription factor STAT3, which forms a dimer that is dependent on p300 acetyl-transfer activity [109]. Inhibition of STAT3 acetylation hampers its DNA-binding capability and therefore affects its dimerization [110]. The combination of the monomer concentration, the PTMs available, and the binding affinity between monomers determine the ratio of dimerization and further function. The main consequences of dimerization may be to increase binding affinity, protein stability, and specificity. Therefore, dimerization in combination with PTMs may potentially explain the differences between the predicted and observed ZEB1 molecular weights. Identifying the modifications that mediate ZEB1 protein stability and whether the PTMs regulate dimerization abilities will require further study by the field, and the results may shed additional light on understanding its molecular associations and function.

Herein, we highlighted the current understanding of the biological function of ZEB1, as well as its post-transcriptional and post-translational regulation. Several questions regarding ZEB1’s regulation by PTMs remain unanswered, and our goal is to further elucidate the novel molecular mechanisms of ZEB1. We identified numerous reports that solidified the role of ZEB1 as a transcriptional repressor and/or activator by the recruitment of cofactors. ZEB1, through the direct action of PTMs, promotes complementary or synergistic effects in some cells but has a seemingly opposite role in others. Studies that identify all of ZEB1’s interacting partners and any PTMs that might modulate its function and molecular associations are required to elucidate ZEB1 protein stability, interaction, and dimerization with an eye towards developing treatment strategies to combat metastasis.

## Figures and Tables

**Figure 1 cancers-14-01864-f001:**
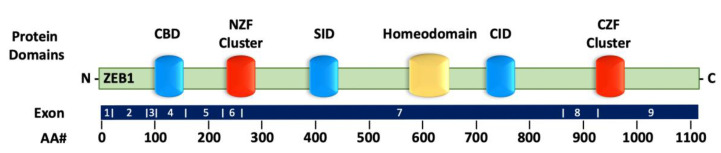
The ZEB1 structure. The homologous structural homeodomain (at aa 581–640) is in between the amino- and carboxy-terminal zinc finger domains (aa 240 to 277 and aa 918 to 971). The CBD, SID, and CID are shown. NZF: amino-terminal zinc finger cluster, CZF: carboxy-terminal zinc finger cluster, SID: Smad interaction domain, CID: CtBP interaction domain, and CBD: P300-P/CAF interaction domain.

**Figure 2 cancers-14-01864-f002:**
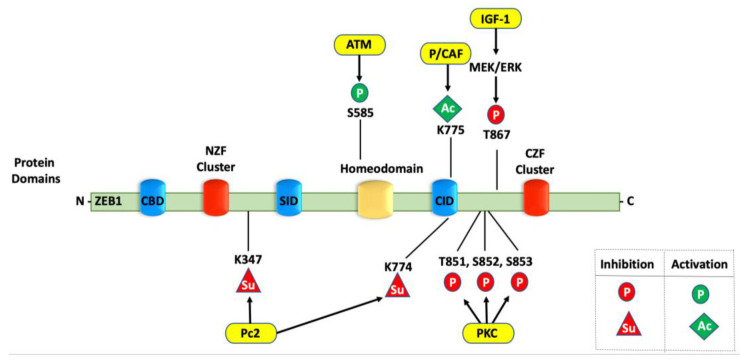
Post-translational modifications controlling ZEB1 function. The ZEB1 protein with its two terminal zinc finger domains (NZF and CZF) and interaction domains (CBD, SID, and CID) is shown. The covalent (i.e., post-translational) modifications that activate or inhibit ZEB1 function are shown in green and red, respectively. The circles indicate phosphorylation (P); the diamond indicates acetylation (Ac); and the triangles indicate sumoylation (Su). The yellow ovals indicate the enzymes catalyzing these modifications.

**Figure 3 cancers-14-01864-f003:**
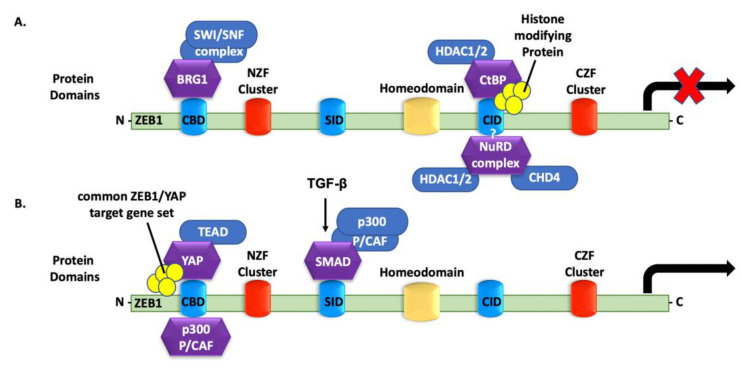
Transcriptional cofactors are required for the ZEB1 function. (**A**) ZEB1 interacts with co-repressors to repress the transcription of their target genes. BRG1 binds to the CBD, and CtBP binds to the CID, but the binding site of the NuRD complex is still unknown; (**B**) ZEB1 interacts with co-activators to activate the transcription of their target genes. p300-P/CAF and YAP bind to the CBD. P300 forms a complex with Smad (Smad-P300) in response to TGF-β via its SID.

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
