# Peer review of "Regulation of ZEB1 Function and Molecular Associations in Tumor Progression and Metastasis"

_cancers, 2022, doi:10.3390/cancers14081864_

Round 1

Reviewer 1 Report

I would like to thank the editor and associated editor for giving me the opportunity to review this article.

The review article is very well written. The authors reviewed the transcriptional and post-translational modifications/regulation of ZEB1, an important EMT transcription factor. It is comprehensive and included all the important publications with an emphasis on PTMs of ZEB1.

The sections are well organized. The language is easy to understand and flows smoothly. There is no grammar error that I can find.

The three figures included are well made. They are simple but clearly demonstrate the points. It is enjoyable to look at.

Overall, I highly recommend accepting and publishing this article. It will contribute to the understanding of ZEB1 and EMT, and give insights into targeted cancer therapy.   

Author Response

We appreciate Reviewer 1 for reviewing this article and the great interest in accepting and publishing this article as submitted.

Reviewer 2 Report

Perez-Oquendo et al., summarized the “regulation of ZEB1 function and molecular association in tumor progression and metastasis”. As ZEB1 plays shows significant upregulation in several tumor types, they provide the transcriptional and post-translational modification of ZEB1 and its association to tumor progression. The authors do a good job in providing a rigorous examination of the literature related to ZEB1 regulation through post-translational modification.

Overall, the review is well summarized, and it will provide novel information to the readers in cancer research field. Some minor corrections need to carry out in the manuscript.

  1. In the section 2.2 (ZEB1 in tumor progression), the authors have only mentioned about the upregulation of ZEB1 in different cancer types. Explain in brief (at least for two -three cancer type) what all the mechanism or signaling cascade through which ZEB1 mediates tumor progression.
  2. Add few points on the role of ZEB1 in therapy-resistance, cancer stem cells and immune infiltration or regulation during tumor pathogenesis.
  3. Emerging studies demonstrate the epigenetic regulation of genes during epithelial-to-mesenchymal transition. Is there any study that describes the role of ZEB1 in mediating epigenetic regulation?
  4. Add few lines about targeting the ZEB1 in cancer. Also add some challenges to target ZEB1

Author Response

We appreciate Reviewer 2 for the constructive comments which we believe have strengthened the manuscript. Please see our responses below:

  1. We have added a new discussion of the mechanisms by which ZEB1 mediates tumor progression in various tumor types in section 2.2.
  2. We have added some of the known roles of ZEB1 in cancer stem cells and tumor immune infiltration in section 2.2. Additionally, we have added section 2.4 to discuss the role of ZEB1 in mediating resistance to therapy.
  3. We have noted and added section 3.3 to discuss the role of ZEB1 in mediating epigenetic regulation.
  4. We have added a discussion of ZEB1-targeting agents in cancer in section 2.4. Furthermore, we noted and added in the conclusions some challenges in targeting this important transcriptional repressor.